# Induction of Superior Systemic and Mucosal Protective Immunity to SARS-CoV-2 by Nasal Administration of a VSV–ΔG–Spike Vaccine

**DOI:** 10.3390/vaccines12050491

**Published:** 2024-05-01

**Authors:** Yfat Yahalom-Ronen, Sharon Melamed, Boaz Politi, Noam Erez, Hadas Tamir, Liat Bar-On, Julia Ryvkin, Dena Leshkowitz, Ofir Israeli, Shay Weiss, Amir Ben-Shmuel, Moria Barlev-Gross, Lilach Cherry Mimran, Hagit Achdout, Nir Paran, Tomer Israely

**Affiliations:** 1Department of Infectious Diseases, Israel Institute for Biological Research, Ness Ziona 74100, Israel; yfatyr@iibr.gov.il (Y.Y.-R.); sharonm@iibr.gov.il (S.M.); boazp@iibr.gov.il (B.P.); noame@iibr.gov.il (N.E.); hadast@iibr.gov.il (H.T.); shayw@iibr.gov.il (S.W.); amirb@iibr.gov.il (A.B.-S.); moriab@iibr.gov.il (M.B.-G.); lilachc@iibr.gov.il (L.C.M.); hagita@iibr.gov.il (H.A.); 2Department of Biochemistry and Molecular Genetics, Israel Institute for Biological Research, Ness Ziona 74100, Israel; liatb@iibr.gov.il (L.B.-O.); ofiri@iibr.gov.il (O.I.); 3Bioinformatics Unit, Life Science Core Facilities, Weizmann Institute of Science, Rehovot 52621, Israel; yulia.ryvkin@weizmann.ac.il (J.R.); dena.leshkowitz@weizmann.ac.il (D.L.)

**Keywords:** SARS-CoV-2, vaccine, nasal vaccination, immunity, VSV–∆G–spike, mucosal vaccination

## Abstract

The emergence of rapidly spreading variants of Severe Acute Respiratory Syndrome Coronavirus 2 (SARS-CoV-2) poses a major challenge to vaccines’ protective efficacy. Intramuscular (IM) vaccine administration induces short-lived immunity but does not prevent infection and transmission. New vaccination strategies are needed to extend the longevity of vaccine protection, induce mucosal and systemic immunity and prevent viral transmission. The intranasal (IN) administration of the VSV–ΔG–spike vaccine candidate directly to mucosal surfaces yielded superior mucosal and systemic immunity at lower vaccine doses. Compared to IM vaccination in the K18–hACE2 model, IN vaccination preferentially induced mucosal IgA and T-cells, reduced the viral load at the site of infection, and ameliorated disease-associated brain gene expression. IN vaccination was protective even one year after administration. As most of the world population has been vaccinated by IM injection, we demonstrate the potential of a heterologous IM + IN vaccination regimen to induce mucosal immunity while maintaining systemic immunity. Furthermore, the IM + IN regimen prevented virus transmission in a golden Syrian hamster co-caging model. Taken together, we show that IN vaccination with VSV–ΔG–spike, either as a homologous IN + IN regimen or as a boost following IM vaccination, has a favorable potential over IM vaccination in inducing efficient mucosal immunity, long-term protection and preventing virus transmission.

## 1. Introduction

Current COVID-19 vaccines, whether mRNA-, viral vector- or protein-based, have significantly contributed to the battle against COVID-19 by providing protection against severe disease and death. With time, emerging variants of SARS-CoV-2 displayed increased infection and spreading capabilities, along with reduced sensitivity to the neutralization by antibodies induced by these vaccines [1,2,3,4,5]. Consequently, SARS-CoV-2 variants continue to spread among vaccinated individuals. Such ongoing transmission and virus replication, especially in immunized individuals, allows for the emergence of new immune-evasive variants that further limit the effectiveness of current vaccines. With more than 3 years into the COVID-19 pandemic and a total of billions of vaccines having been administered intramuscularly, a large proportion of the world’s population has prior immunity provided by intramuscular (IM) vaccination. An IM injection induces preferentially systemic immune responses, but it does not evoke significant mucosal immunity, a valuable tool in confronting respiratory-borne agents like the current circulating variants of SARS-CoV-2. Mucosal immunity can be achieved by introducing the vaccine through the respiratory tissues. Vaccines that induce an effective and durable mucosal immunity in the respiratory tract have the potential to prevent virus infection and replication, and consequently prevent shedding and spreading, and may reduce the rate of variant emergence [6,7]. Improved vaccine efficacy may allow for dose sparing, thus reducing the cost of vaccine production. Additionally, nasal administration is not invasive, causes little discomfort and might reduce the rate of vaccine hesitancy [8]. There are global efforts to develop IN vaccines against SARS-CoV-2, some of which are already in clinical trials. Due to the unique characteristics of the mucosal environment, mucosal vaccines for SARS-CoV-2 have been developed based on viral vectors or proteins [9], many of which are exploring one or two IN doses or combined IM and IN administration [9,10,11,12,13,14]. A recent review by Kinsely et al. further discussed mucosal vaccines’ potential and benefits, as well as the challenges [6].

We have previously developed a VSV–ΔG–spike vaccine candidate based on the vesicular stomatitis virus (VSV) platform, in which the VSV glycoprotein G (VSV–G) was replaced by the full-length human codon-optimized S gene encoding the SARS-CoV-2 spike glycoprotein (S) [15]. The VSV platform has been successfully used for other vaccines such as the FDA-approved rVSVΔG-ZEBOV-GP Ebola vaccine [16] (Ervebo^®^, Merck Sharp & Dohme LLC Rahway, NJ, USA) and rVSV∆G-LASV-GPC for Lassa virus [17], which is currently in phase II clinical trials. During the VSV–ΔG–spike vaccine propagation process by serial passaging in cultured cells, several spontaneous mutations emerge, which are identical or correspond with mutations that are known to characterize several SARS-CoV-2 variants and contribute to its unique sequence. These include mutations at N501, E484, Q493, and G685 of the spike protein [18,19,20,21]. We have also demonstrated the nonclinical safety and immunogenicity of the VSV–ΔG–spike vaccine in four animal species: mice, hamsters, rabbits and pigs [22,23]. Neurovirulence was tested by an intracranial injection (IC) administered to both immune-competent C57BL/6 mice and sensitive type-I interferon knock-out mice (Ifnar-/-), resulting in no signs of morbidity or body weight loss. The ability of the vaccine to propagate at the site of injection following IM administration was also evaluated in K18–hACE2 transgenic mice, expressing the human ACE2 receptor under the keratin 18 promoter (K18) in epithelial cells in various tissues, including airway epithelia, namely the route of SARS-CoV-2 infection [21,24,25,26,27]. A gradual clearance of the VSV–ΔG–spike vaccine at the site of injection was demonstrated, and virus in the lungs was not detected at any time point [22]. A nonclinical repeated-dose (GLP) toxicity study in rabbits revealed only restricted treatment-related changes, which correlated with a trend of recovery [23]. Altogether, it was concluded that the VSV–ΔG–spike vaccine is safe, and these results further supported the initiation of clinical trials (NCT04608305). We have also addressed vaccine efficacy and shown that the VSV–ΔG–spike vaccine induces neutralizing antibodies in mice and humans that efficiently neutralize the original virus, as well as the alpha, beta, gamma, delta [21] and omicron variants [20], and protects hamsters and K18–hACE2 transgenic mice against challenges with various SARS-CoV-2 variants [15,20,21].

VSV–ΔG–spike as a replicating live vaccine exerts its functions both as an antigen, presenting spikes on the surface of the vaccine virus, and as a live virus, expressing the spike protein in hACE2-expressing cells infected by VSV–ΔG–spike following vaccination. Human ACE2 (hACE2), the major receptor mediating SARS-CoV-2 entry into cells [28], is mainly expressed and presented on the surface of alveolar type II epithelial cells (ATII), decorating the respiratory and digestive tracts [29]. Thus, IN vaccination with VSV–ΔG–spike has the potential to induce robust mucosal, as well as systemic, anti-spike immunity.

Mucosal immunity, similarly to systemic immunity, is a combined action of innate and adaptive components, yet each system has unique characteristics. The mucosal immune system provides protection already at the site of infection—the respiratory tract. Mucosal adaptive immunity, and specifically that of the respiratory organs, is composed of several components, including secretory immunoglobulin A (sIgA) and tissue-resident memory CD8+ T-cells (T_RM_). Secretory IgA can neutralize viral particles at the mucosal surface before the infection of epithelial cells takes place [30] and can protect against a broader spectrum of pathogens at the mucosa [30]. Not only does the site of vaccination affect the consequent immune response, but so does the type of vaccine used. Indeed, local antigen deposition by mucosal vaccination (as a protein, virus particle or mRNA) is key for the induction of mucosal IgA [31].

To mitigate the need for improved vaccines capable of inducing efficient and increased breadth and durable systemic and mucosal immunity, we chose to evaluate VSV–ΔG–spike mucosal vaccination. To this end, we utilized the K18–hACE2 transgenic mouse model [21,24,25,26,27]. SARS-CoV-2 infection of K18–hACE2 mice was previously shown to target the brain as a major site, leading to neuroinflammation, neurological disease, morbidity and mortality [32,33]. Infected mouse brains were reported to display a pathology similar to the clinical symptoms of patients with severe COVID-19 disease, further supporting K18–hACE2 as a model that recapitulates SARS-CoV-2 infection in humans [25]. We compared IM injection to IN instillation through the characterization of the local and systemic immune responses and their protection efficacy against lethal challenges. We also evaluated the added value of IN vaccination following IM vaccination (heterologous IM + IN prime and boost regimen) since a large proportion of the world’s population has previously been IM vaccinated. We also addressed the durability of protective mucosal immunity. In light of our previous work showing cytokine storm and immune activation in SARS-CoV-2 variant-infected mouse brains, we performed a transcriptomic analysis of K18–hACE2 mouse brains infected with SARS-CoV-2 following IN instillation to assess the vaccine’s impact on the host response. Moreover, to evaluate the ability of mucosal vaccination with VSV–ΔG–spike to prevent SARS-CoV-2’s transmissibility and spreading, we applied an additional animal model, namely a golden Syrian hamster model, in a co-caging setting.

Altogether, our data demonstrate the potential of mucosal vaccination with VSV–ΔG–spike, either in IN + IN administration or following IM priming, in the prevention of disease and death while minimizing the viral load at the respiratory organs, preventing virus spread and inducing long-term immunity and protection.

## 2. Methods

### 2.1. Cells

Vero E6 cells (ATCC CRL-1586™) were grown in DMEM containing 10% fetal bovine serum (FBS), MEM non-essential amino acids (NEAA), 2 mM L-glutamine, 100 Units/mL penicillin, 0.1 mg/mL streptomycin, and 12.5 Units/mL nystatin (P/S/N). All reagents were from Biological Industries, Beit-Haemek, Israel. Cells were cultured at 37 °C, 5% CO_2_ and 95% humidity.

### 2.2. Viruses

Virus stocks of the SARS-CoV-2 original virus (GISAID accession EPI_ISL_406862) were propagated in Vero E6 cells (by 4 passages). The SARS-CoV-2 delta variant (B.1.617.2, GISAID accession EPI_ISL_4169986) was provided by the Central Virology Lab of the Israel Ministry of Health, propagated in Vero E6 cells, and verified by whole-genome sequencing (WGS, [20]). Both virus stocks were titered on Vero E6 cells, as previously described [15]. Handling and working with the SARS-CoV-2 virus were conducted in a BSL3 facility in accordance with the biosafety guidelines of the Israel Institute for Biological Research (IIBR).

### 2.3. Animal Experiments

All animal studies were approved by the Israel Institute for Biological Research institutional animal care and use committee (IACUC). All animal experiments involving SARS-CoV-2 were conducted in a BSL3 facility. Six-to-eight-week-old female K18–hACE2 transgenic mice (The Jackson Laboratory, Bar Harbor, ME, USA) and 6–7-week-old golden Syrian hamsters were used (Charles River Laboratories, Kingston, NY, USA). The animals were maintained as previously described [21]. Both mice and hamsters were randomly assigned to the experimental groups.

The vaccination of K18–hACE2 mice with the VSV–ΔG–spike vaccine was performed either intramuscularly (IM) or intranasally (IN) in a homologous (IM + IM, or IN + IN) or heterologous (IM + IN) prime-boost regimen at varying vaccine doses ranging from 10^3^ to 10^7^ pfu per animal. Intramuscular vaccination was performed through the injection of 50 µL of the VSV–ΔG–spike vaccine in a phosphate buffered saline (PBS) solution supplemented with 2% heat-inactivated FBS into the thigh muscle. IN vaccination was performed through the administration of 20 µL of the VSV–ΔG–spike vaccine in a phosphate buffered saline supplemented with 2% heat-inactivated FBS into the nostrils of anesthetized mice (75 mg/kg ketamine, 7.5 mg/kg xylazine in PBS). Vaccination was given in a prime and boost regimen with a three-week interval.

Serum samples for ELISA and PRNT_50_ were obtained approximately 7–14 days post-boost vaccination for antibody testing. Bronchoalveolar lavage fluids (BALFs) were obtained 1 week following the boost vaccination.

For animal challenge, the SARS-CoV-2 virus was diluted in PBS supplemented with 2% FBS to the appropriate titer.

Approximately 4 weeks following the boost vaccination, anesthetized K18–hACE2 mice (75 mg/kg ketamine, 7.5 mg/kg xylazine in PBS) were infected into both nostrils with an IN instillation (20 µL/mouse) with 10,000 pfu of the delta variant of SARS-CoV-2. Mice were monitored daily for body weight changes. Mice were sacrificed to assess the viral load in the lungs (at 2 and 5 dpi), nasal turbinates (2 dpi) and brains (5 dpi), and for a gene expression analysis of the brains (4 dpi). All analyses are detailed below.

Hamsters were vaccinated with the VSV–ΔG–spike vaccine by a prime IM injection with 10^6^ pfu followed by 20 µL of an IN boost of 10^3^ pfu 21 days later, to intraperitoneal anesthetized hamsters (IP, 115 mg/kg ketamine, 4.5 mg/kg xylazine in PBS). Fourteen days post-boost vaccination, all vaccinated and unvaccinated hamsters were anesthetized by IP injection and challenged by IN instillation (50 µL/hamster) with 5 × 10^6^ pfu of the original SARS-CoV-2 into both nostrils. Hamsters were monitored daily for body weight changes. Viral load analysis of the lungs or nasal turbinates at 3 dpi was performed as detailed below.

### 2.4. Co-Caging of Hamsters

Hamsters were vaccinated with the VSV–ΔG–spike vaccine by prime IM injection followed by a boost IN instillation and challenged 14 days later, as described above. These challenged hamsters will be defined from here on as “donors” (either vaccinated donors or unvaccinated donors). Three days post-challenge, each donor was co-caged with two naïve, unvaccinated hamsters (“acceptors”). The co-caging setting included a cage, which was separated into two compartments by a semi-permeable 0.5 × 8 cm metal net, allowing for uninterrupted air flow and limited contact (i.e., nasal/mouth or saliva drop contact). Twenty-four hours post-co-caging, donors and acceptors were separated, donors were sacrificed (4 dpi) and the viral load was determined at both the nasal turbinates and lungs. Three days post-co-caging, acceptors were sacrificed, and the viral load was determined at both the nasal turbinates and lungs.

### 2.5. Tissue Processing for BALFs and Flow Cytometry

For tissue processing, mice were euthanized by IP injection of Pentobarbiton (300 mg/kg). Bronchoalveolar lavage fluids (BALFs) were collected through the insertion of an 18 G catheter into the trachea and washing with 1 mL of PBS. The lungs and spleens were cut and digested into single-cell suspension, as previously described [34].

### 2.6. Flow Cytometry

All washing steps were performed using FACS buffer (PBS + 0.5% FBS + 2 nM EDTA). Tetramer staining was performed as previously described [34]. Samples were acquired on LSRFortessa (BD Biosciences, San Jose, CA, USA) and analyzed with FlowJo V.10 software (TreeStar, Ashland, OR, USA). The following mAb clones were used for staining: CD3 (145-2C11), CD8 (53–6.7), CD4 (GK1.5) and CD62L (MEL-14). All antibodies were purchased from Thermo Fisher Scientific or BD Biosciences (San Jose, CA, USA). H-2K(b) SARS-CoV-2 S539-546 (VNFNFNGL). Tetramers were provided by the NIH Tetramer Core Facility (Atlanta, GA, USA).

### 2.7. ELISpot Assay

The detection of IFNγ-secreting cells was performed as previously described [34,35]. Briefly, 4 × 10^5^ splenocytes or lung cells (processed as described above) were plated into anti-IFNγ pre-coated 96-well ELISpot plates and incubated for 24 h at 37 °C in the presence of the immunodominant MHC class I H-2K(b)-restricted epitope of the spike amino acids 539–546 VNFNFNGL (S539) at a final concentration of 2 µg/mL. The visualization of IFNγ-secreting cells was performed using a Mouse IFN-γ single-color ELISpot kit (Cellular Technology Limited, Biotec, Bonn, Germany). The quantification of cytokine-secreting cells was performed as previously described [34]. Antigen-free cells supplemented with medium were used as a negative control.

### 2.8. Enzyme-Linked Immunosorbent Assay (ELISA)

ELISA was performed as previously described [36]. Briefly, plates were coated with 100 ng/mL of recombinant SARS-CoV-2 S glycoprotein (S2P, [37]) in carbonate bicarbonate at 4 °C overnight. Naive or vaccinated mouse sera were used at a dilution of 1:100 in TSTA. Anti-mouse IgG– or IgA–alkaline phosphatase (AP) conjugates were diluted to 1:1000 and used as secondary antibodies. *P-*nitrophenyl phosphate (pNPP) substrate (Sigma, St. Louis, MO, USA) was added after 3 washes, and the optical density was measured (OD of 405 nm). IgG or IgA values were determined by subtracting the value of blank wells plus 3XSTDEV from each sample.

### 2.9. Plaque Reduction Neutralization Test (PRNT_50_)

A plaque reduction neutralization test (PRNT_50_) was performed on Vero E6 cells, as previously described [21], using sera from vaccinated mice obtained at approximately 2 weeks following the boost. Sera were diluted in 300 µL and mixed with 300 µL of 300 pfu/mL of the delta variant of SARS-CoV-2 and incubated for 1 h, and the cells were infected with a serum–virus mixture (200 µL/well) and incubated at 37 °C and 5% CO_2_ for 1 h. Overlay (MEM containing 2% FBS and 0.4% tragacanth (Merck, Rehovot, Israel)) were added to each well, and the plates were incubated for 72 h. Following incubation, cells were fixed and stained. NT50 was calculated using Prism 6 software (GraphPad Software Inc., San Diego, CA, USA).

### 2.10. Viral Load Determination in Organs

For the K18–hACE2 mice, the viral load was determined for the nasal turbinates (2 dpi), lungs (2 and 5 dpi) and brains (5 dpi). For the hamsters, the viral load was determined for the nasal turbinates and lungs (3 or 4 dpi). Organs were harvested and processed, and the quantitation of infectious virus was performed with a plaque assay on Vero E6 cells, as previously described [15]. The number of plaques in each well was determined. The viral load, as well as the limit of detection (LOD), were calculated based on the volume of cell infection (200 µL/well), dilution factor, and tissue processing volume: for mice: 1 mL for nasal turbinates and 1.5 mL for lungs and brains; and for hamsters: 2 mL for nasal turbinates and 3 mL for lungs and brains. Viral loads are presented as pfu/organ.

### 2.11. Gene Expression and Functional Analysis

RNA was isolated from the K18–hACE2 mouse brains at 4 dpi using a TRIzol reagent (Invitrogen, Carlsbad, CA, USA). RNA quantification was performed in a Qubit fluorometer using the Qubit RNA HS assay kit (Invitrogen, Carlsbad, CA, USA). The quality control analysis of RNA integrity was performed using High-Sensitivity RNA ScreenTape and the TapeStation Analysis software (Agilent Technologies, Santa Clara, CA, USA). The RIN (RNA integrity number) scores of the samples were > 7. RNAseq was performed at the JP Sulzberger Columbia Genome Center (New York, NY, USA). Libraries were generated using the Illumina TruSeq stranded mRNA kit (San Diego, CA, USA). Polyadenylated RNA enrichment was performed. The sequencing of the 100-bp paired-end reads was performed on the Illumina NovaSeq 6000 system. Sequencing data were analyzed using UTAP 1.10.2 [38]. Reads were trimmed using cutadapt [39] (parameters: -a AGATCGGAAGAGCACACGTCTGAACTCCAGTCAC -a “A{10}” -a “T{10}” -A “A{10}” -A “T{10}” –times 2 -q 20 -m 25). Reads were mapped to mouse genome (mm10) using STAR [40] v2.5.2b. (parameters: –alignEndsType EndToEnd, –outFilterMismatchNoverLmax 0.05, –twopassMode Basic). Gene annotations from RefSeq were used to quantify gene expression. Counting was carried out using STAR. Further analysis was performed for genes with a minimum of 5 reads in at] least one sample. The normalization of the counts and differential expression analysis were performed using DESeq2 [41] with the following parameters: betaPrior = True, cooksCutoff = FALSE, independentFiltering = FALSE. Raw *p* values were adjusted for multiple testing using the procedure of Benjamini and Hochberg [42]. Differential gene expression was calculated by a comparison of the data of the virally infected mice with the naïve mice. The criteria for significance were as follows: padj ≤ 0.05, |Log2FoldChange| ≥ 1, BaseMean ≥ 5. The log2 Fold Change (Log2FC) was converted to Fold Change values in some cases for simplicity.

Area-proportional Venn diagrams were created using the eulerr_7.0.2 R package [43].

A heatmap of the scaled z-score and rlog-transformed normalized read counts was created using Seaborn, Python, Version 0.12.2. [44]. The top 30 genes were selected based on the unvaccinated group’s Fold Change score.

Canonical pathway analysis identified enriched pathways using the QIAGEN Ingenuity Pathway Analysis (Version 01-22-01). Differentially expressed genes from the dataset that were associated with a canonical pathway in the QIAGEN Knowledge Base were considered for the analysis. The association significance between the dataset and the canonical pathway was measured by (1) the ratio of the number of molecules from the dataset that map to the pathway divided by the total number of molecules that map to the canonical pathway; and (2) a right-tailed Fisher’s exact test followed by a Benjamini–Hochberg multiple-comparisons correction test. The z-score was calculated to indicate the likelihood of the activation or inhibition of that pathway.

### 2.12. Statistical Analysis

Data were analyzed using GraphPad Prism 6 software. The determination of the statistical significance of anti-spike IgG and anti-spike IgA and the viral load analyses in K18–hACE2 were performed by a one-way ANOVA with a post hoc Tukey’s test. Significance of PRNT_50_, T-cell analyses and viral load analyses in hamsters were performed by a Mann–Whitney nonparametric *t*-test. Statistical analysis for mortality was performed using a log-rank (Mantel–Cox) test. The significance of the association between the dataset and specific canonical pathways was assessed using a right-tailed Fisher’s exact test. A Benjamini–Hochberg multiple-comparisons correction test was applied.

## 3. Results

### 3.1. Induction of Humoral Immune Response by Homologous or Heterologous IM and IN Routes of VSV–ΔG–Spike Vaccination

To evaluate the efficacy of intranasal (IN) immunization in inducing mucosal immunity and protection, we first evaluated the immune response following vaccination with VSV–ΔG–spike in the K18–hACE2 transgenic mice. The mice were vaccinated with either an IM vaccination or an IN instillation at varying doses in a homologous prime-boost regimen with a 3-week interval. To simulate the mucosal vaccination of individuals who have previously received an IM vaccination, we also tested a heterologous route of IM prime vaccination, followed by an IN boost (IM + IN) (Figure 1a). First, we analyzed the serum anti-spike IgG levels of the vaccinated mice following the boost. The spike-specific IgG response was similar or significantly higher following the IN + IN vaccination than following the IM + IM or IM + IN vaccination (Figure 1b). Upon IN + IN vaccination, the antibody titers were not significantly affected by raising the vaccine dose at the tested doses (10^3^–10^5^ pfu). In the mice that were primed with an IM injection with 10^5^ or 10^6^ pfu, IgG titers to spike were not significantly different whether IM or IN boost was administered (Figure 1b, Appendix A). Notably, vaccine-induced immunity was also already observed after the prime vaccination (Appendix A).

Next, the levels of anti-spike IgA, a hallmark of mucosal immune response, were measured. Following homologous IM + IM or heterologous IM + IN vaccination, anti-spike IgA antibodies were induced, with an observed increase following a 10^6^ pfu dose, though not significant (Figure 1c). Similar anti-spike IgA levels were observed for the IM + IM and IM + IN routes, per vaccination dose. In contrast, the IN + IN instillation efficiently induced anti-spike IgA levels that were significantly higher than those induced by the IM + IM vaccination, even at a low dose of 10^4^ pfu (Figure 1c, Appendix A). To assess the protective potential of the above-mentioned doses and regimens, the unvaccinated or vaccinated K18–hACE2 mice were challenged with an IN instillation with a previously determined lethal dose of the SARS-CoV-2 authentic delta variant (100 IN LD_50_, 10,000 pfu, [21]. We show that whereas the IM + IM or IM + IN vaccination required at least 10^5^ pfu in order to provide full protection, a lower dose of 10^4^ pfu was sufficient to provide full protection when IN + IN was administered (Appendix A). Based on the IgG and IgA responses and protection assays, we selected the following vaccination doses for our next experiments: 10^6^ homologous prime and boost IM + IM vaccination, 10^6^ heterologous prime and boost IM + IN vaccination, and 10^4^ homologous prime and boost IN + IN vaccination. The neutralization capacity of the mouse sera following vaccination in these chosen regimens was determined against the authentic delta SARS-CoV-2 variant (Figure 1d). We showed that all the selected regimens efficiently induced neutralizing antibodies to SARS-CoV-2. Importantly, reducing the vaccination dose by 2 logs to 10^4^ pfu in the homologous IN + IN vaccination maintained neutralization potential, which was significantly higher than following the homologous IM + IM 10^6^ pfu vaccination regimen (Figure 1d).

### 3.2. Induction of Mucosal Immunity in BALFs by Mucosal Vaccination

To further characterize the mucosal immune response, we determined the level of anti-spike antibodies in the bronchoalveolar lavage fluids (BALFs). The K18–hACE2 mice were vaccinated with 10^6^ pfu of IM + IM or IM + IN and with 10^4^ pfu of IN + IN in a 21-day interval, and BALFs were collected 7 days post-boost vaccination. The anti-spike IgG antibodies in the BALFs were significantly higher following the IN + IN regimen than when following either the IM + IM or IM + IN vaccination regimens, despite the fact that the IN + IN regimen was carried out at a lower dose of 10^4^ pfu, whereas the IM + IM or IM + IN regimens were carried out at a 10^6^ pfu dose (Figure 1e). The IM + IM and IM + IN regimens induced similar anti-spike IgG levels. The anti-spike IgA antibodies in the BALFs were induced only in mice receiving at least one IN vaccination, i.e., the IM + IN and IN + IN regimens (Figure 1f). Interestingly, the IgA levels in the BALFs were significantly higher in the IM + IN group than in the IN + IN group. Altogether, our results demonstrate the added value of a mucosal vaccination for the induction of BALF-associated IgA antibodies. Taken together, we showed the induction of humoral immunity in both sera and BALFs by incorporating IN vaccination in the vaccination regimen, even when a low vaccination dose was used.

### 3.3. Induction of Spike-Specific T-Cell Response Following Mucosal Vaccination

To better characterize the immune response following IN versus IM vaccination with VSV–ΔG–spike, the K18–hACE2 mice were vaccinated with the IM + IM, IM + IN or IN + IN regimen, as described above. Seven days post-boost, the animals were sacrificed, and the lungs and spleens were collected for an evaluation of the T-cell response. All the vaccination regimens induced both a systemic (spleen; Figure 2a) and local (lung; Figure 2b) CD8+ T-cell response, which was depicted by an increase in spike-specific CD8 T-cells in comparison to the unvaccinated animals. However, the IM + IN vaccination induced significantly higher numbers of specific T-cells in comparison to the other vaccination regimens. A flow cytometry analysis further demonstrated that the IM + IN vaccination induced a significantly higher frequency of spike-specific effector T-cells in the lungs in comparison to the IM + IM or IN + IN regimen, as represented by the CD62L^low^, tetramer-positive population (Figure 2c,d).

### 3.4. Protection of Vaccinated Mice against SARS-CoV-2 Challenge

Having demonstrated that mucosal vaccination induces both systemic and mucosal humoral and cellular immunity, we further evaluated the protective efficacy conferred by the different vaccination regimens. Thus, mice were vaccinated in the IM + IM or IM + IN regimen with 10^6^ pfu, or the IN + IN regimen with 10^4^ pfu, as described above. Fourteen days post-boost, the mice were challenged by an IN instillation with a lethal dose of the SARS-CoV-2 delta variant (100 IN LD_50_, 10,000 pfu). As expected, unvaccinated mice started to lose weight 3 days post-infection (dpi), gradually deteriorated and succumbed to the infection within 6–9 days. In contrast, all the vaccinated mice did not show any signs of morbidity (Figure 3a) and survived the lethal challenge (Figure 3b).

Next, we evaluated the vaccine’s protective effect on the SARS-CoV-2 viral load in the target organs with an emphasis on the nasal turbinates (NTs), lungs and brain. The unvaccinated or vaccinated mice in the above-mentioned regimens and doses were similarly challenged with the SARS-CoV-2 delta variant. At 2 or 5 dpi, the target organs were removed and analyzed by a plaque assay to determine the infectious virus load in the NTs and lungs (2 dpi, Figure 3c,d) or lungs and brains (5 dpi, Figure 3e,f). As expected, all the unvaccinated mice displayed high viral titers in all the tested organs. In contrast, all the vaccinated mice had significantly lower viral loads in all the tested organs. We also observed a reduction in the viral load of the animals that were vaccinated at least once intranasally (IM + IN, or IN + IN) in comparison to muscular vaccination; however, this reduction was not statistically significant. Moreover, in three out of four IN + IN-vaccinated mice, the viral titers in the lungs were below the limit of detection (LOD, Figure 3d). At 5 days post-infection, the infectious virus was undetectable in both the lungs and the brain of all the vaccinated mice, indicating a reduction in the viral load of more than three or seven orders of magnitude, respectively (Figure 3e,f). Altogether, whereas all the vaccination routes led to a significant reduction in the viral load in the target organs, a trend towards reduced viral loads in the respiratory organs was observed in the mice vaccinated by an IN instillation.

### 3.5. Prevention of Altered Brain Gene Expression by Mucosal Vaccination

To further elaborate on the cellular pathways associated with immune protection or disease and mortality, we performed an RNAseq analysis of total RNA samples from whole brains of unvaccinated mice, or mice vaccinated with the IM + IM, IM + IN or IN + IN regimen at 4 days post-infection with the variant delta of SARS-CoV-2. As described, in K18–hACE2 mice, the brain is a major target site for SARS-CoV-2, as also expressed by their high brain viral titers. Our previous work demonstrated that the infection of K18–hACE2 mice with several SARS-CoV-2 variants (original virus, alpha, beta, gamma and delta) resulted in the expression of viral antigens in several brain areas, as well as the induction of neuroinflammation- and cytokine storm-related genes [21]. Hence, we focused on the brain for the gene expression analysis. Following infection, a total of 2289 genes were significantly altered in the unvaccinated and infected mouse brains compared to the naïve mouse brains (unvaccinated and uninfected) (Log_2_FC > 1, or < −1, padj ≤ 0.05). The total numbers of differentially expressed genes (DEGs) in the IM + IM, IM + IN or IN + IN groups following infection were lower: 711, 1412 and 1598 DEGs, respectively (Figure 4a). More than half of the unvaccinated group’s DEGs (1515) were also significantly altered in the vaccinated groups. In the control unvaccinated and infected brains, a little over 50% of the DEGs were upregulated (1181 DEGs, Figure 4b,c). The IM + IM group displayed the lowest number of DEGs of all the vaccinated groups, and only one-tenth of them (67 DEGs) were unique to this group and did not overlap with any of the other groups (Figure 4d). In the IM + IN and IN + IN groups, about one-third of the DEGs were unique (378 and 537 DEGs, respectively, Figure 4d). Also, in each of the vaccination regimens, most of the DEGs were downregulated (572, 759 and 877 DEGs in IM + IM, IM + IN and IN + IN, respectively, Figure 4d–f), and most of them overlapped (450 DEGs, Figure 4f). Overall, the number of DEGs in the brains of the unvaccinated–infected mice was higher than the number of DEGs in the vaccinated–infected groups. Also, the unvaccinated–infected group’s DEGs were mostly upregulated, as opposed to the mostly downregulated DEGs in the vaccinated–infected mice’s brains.

We next focused on the top 30 DEGs of the unvaccinated group (padj ≤ 0.05, arranged by Log2FC) and tested their expression in the vaccinated–infected groups. We showed that for the unvaccinated and infected brains, many of the top 30 genes were members of the CCL and CXCL families (Figure 4g). In the vaccinated and infected groups, most of these 30 genes were non-significant; thus, they are not differentially expressed relative to the naïve mice’s genes. One animal in the IM + IN group showed activation of most of the DEGs activated by the unvaccinated group, but to a lesser degree. Importantly, in the IM + IM- and IN + IN-vaccinated and infected groups, only a few genes were differentially expressed. Moreover, the log2FC values for most of the DEGs in the vaccinated and infected groups were lower compared to the unvaccinated and infected brain DEGs (Figure 4g).

To further explore the biological pathways that were affected upon infection of vaccinated vs. unvaccinated mouse brains, we performed a functional enrichment analysis using Ingenuity Pathway Analysis software (IPA, Version 01-22-01). First, we compared the DEG-enriched pathways of the brains of the unvaccinated mice and those following the three vaccination regimens at 4 days following delta SARS-CoV-2 infection and focused on five signaling categories: pathogen-influenced signaling, cytokine signaling, disease-specific pathways, humoral immune response and cellular immune response. An enrichment of most of the pathways was observed in the unvaccinated group (Figure 5a and Appendix A), while both homologous vaccination groups (IM + IM and IN + IN) did not result in enrichment of many pathways (Figure 5, Appendix A and Appendix A). One of the most enriched (by Benjamini and Hochberg (B-H) *p*-value) and activated (by z-score) pathways in the brains of the unvaccinated mice was “Role of hypercytokinemia/hyperchemokinemia in pathogenesis of influenza” (Figure 5a). This pathway was also enriched and activated in the brains of IM + IN-vaccinated mice, but to a much lesser degree (Figure 5b). An additional enriched and activated pathway in the unvaccinated mice’s brains was the “Pathogen-induced cytokine storm signaling pathway” (Figure 5a). This pathway is comprised of members of the CCL family, members of the CXCL family, such as CXCL10, CXCL9, IRF1, IRF7, IRF9, IL15, IL18, IL1a and IL1b, and many more. This pathway was not enriched in any of the vaccinated–infected groups. (Figure 5 and Appendix A). In the IM + IN vaccinated and infected groups, we also observed an enrichment of the “cellular immune response”-associated pathways, such as the “Th1 pathways” and “Th1 and Th2 activation pathway” (Figure 5b). Interestingly, the pathway analysis detected no significant pathway enrichment in the IM + IM vaccinated–infected group and only three enriched pathways in the IN + IN vaccinated–infected group, which categorized into “Disease-Specific Pathways” and “Cellular Immune Response” among the five above-mentioned categories (Appendix A). The three pathways “coronavirus pathogenesis pathway”, “Granzyme A signaling” and “Mitochondrial Dysfunction” all presented negative z-scores. Altogether, the gene expression profiling highlights the profound difference between the robust activation of multiple pathways in the brains of the unvaccinated mice, while the vaccinated mice displayed mostly silent or suppressed gene expression and, in some cases, the activation of immune pathways.

### 3.6. Long-Lasting Immunity and Protection Are Achieved by IN Vaccination

In light of the challenging goal of achieving long-lasting immunity to SARS-CoV-2, we performed a small-scale experiment to evaluate vaccine-induced immunity to SARS-CoV-2 at one year following vaccination, as well as the protection of mice against the lethal SARS-CoV-2 challenge (Figure 6a). To this end, mice were vaccinated with prime and boost vaccinations at 3-week intervals by either IM + IM, IM + IN or IN + IN. One year post-vaccination, mice sera were collected, and antibody titers against SARS-CoV-2 spike antigen were determined. We could observe anti-spike IgG antibodies in all tested groups (Figure 6b). We also detected anti-spike IgA antibodies in the vaccinated mice’s sera, with the highest levels in the IN + IN group (Figure 6c). Upon challenge with a lethal dose of SARS-CoV-2 delta variant, all unvaccinated mice deteriorated and succumbed to infection (Figure 6d). Fifty percent of the IM + IM-vaccinated mice survived the lethal delta SARS-CoV-2 infection. The mice that were vaccinated with the IN + IN or IM + IN regimen were completely protected from morbidity and death, further highlighting the preferred efficacy of nasal vaccination with VSV–∆G–spike (Figure 6e).

### 3.7. Prevention of SARS-CoV-2 Transmission by IN Vaccination in a Hamster Model

So far, we have demonstrated the efficacy of mucosal vaccination in preventing morbidity and mortality and reducing the viral load in the respiratory mucosa in a mouse model. To explore the potential of our vaccine to reduce SARS-CoV-2 transmission by mucosal immunization, we applied it to the golden Syrian hamster model. This model is a well-established COVID-19 disease model that additionally enables the simulation of virus transmission in a co-caging experimental setting [12,13,45]. We chose to explore the IM + IN regimen in order to simulate the IN vaccination of a previously IM-vaccinated population. We first tested the vaccine’s efficacy in preventing disease and reducing the viral load in the lungs and nasal turbinates. To this end, the hamsters were primed with an IM injection of VSV–ΔG–spike and received an IN boost 21 days later. Fourteen days following the boost, both the vaccinated and unvaccinated hamsters were challenged by an IN instillation with 5 × 10^6^ pfu of SARS-CoV-2 and monitored daily for body weight changes as a measure of morbidity (Figure 7a). Additional groups were sacrificed at 3 days post-infection in order to determine the viral load in the NTs and lungs (Figure 7b,c). The unvaccinated hamsters gradually lost about 10% of their initial body weight during the first six days post-infection and then started to gain weight until reaching their initial body weight at about 12 days post-challenge (Figure 7a). In contrast, the vaccinated hamsters were significantly less morbid (days 4–6 post-infection); they lost only about 5% of their initial weight within the first 72 h post-challenge but then gained weight, reaching their initial body weight by day 8 post-challenge (Figure 7a). The analysis of the viral load in the respiratory organs at 3 days post-challenge revealed that while the viral load in the NTs and lungs of the unvaccinated hamsters was higher than 10^6^ pfu per organ (Figure 7b,c), the viral load in the organs of the vaccinated hamsters was significantly lower, by more than three orders of magnitude. Having demonstrated that the vaccination significantly reduced the viral load in both the lower and upper respiratory organs, we asked whether a similar vaccination prior to challenge would also prevent virus transmission to naive hamsters.

To experimentally model virus transmission, we applied a co-caging approach (Figure 7d). The co-caging setting includes a cage which is separated into two compartments by a semi-permeable metal net (0.5 × 8 cm mesh), allowing for uninterrupted air flow and enabling limited contact (contact of nasal, mouth, or saliva drops). The donor hamsters were either unvaccinated or vaccinated, whereas all the acceptors were naive (Figure 7d). Fourteen days post-booster (IN) vaccination, all the donor hamsters, both vaccinated and unvaccinated, were exposed IN to 5 × 10^6^ pfu of SARS-CoV-2. Three days post-exposure, each exposed donor was co-caged (cc) with two naive acceptor hamsters for 24 h, and then the donors and acceptors were separated (Figure 7d). At the time of separation (4 dpi, 1 days post-co-caging (dpcc)), the donor hamsters were euthanized and sacrificed, and the viral load in both the nasal turbinates and lungs was determined (Figure 7e,f for unvaccinated donors; g,h for vaccinated donors). Three days post the co-caging starting point, the acceptors were sacrificed as well, and the viral load was determined in both the nasal turbinates and lungs (Figure 7e,f for acceptors co-caged with unvaccinated donors; g,h for acceptors co-caged with vaccinated donors). All the unvaccinated–infected donors exhibited a viral load of 10^6^ pfu in their NTs and 10^7^ pfu in their lungs (Figure 7e,f, respectively). As expected, most acceptors that were co-caged with unvaccinated–infected donors had a measurable viral load in their NTs and lungs at varying levels, with some organs reaching high loads similar to the loads measured in the donor organs (Figure 7e,f). In contrast, the vaccinated donors had undetectable virus titers in both the NTs and lungs (Figure 7g,h), and consequently, their co-caged acceptors had undetectable viral loads in both organs. Altogether, the co-caging experiment demonstrated the efficacy of IN vaccination with the VSV–∆G–spike vaccine in preventing SARS-CoV-2 transmission and subsequent disease.

## 4. Discussion

The emergence of SARS-CoV-2 led to global efforts to develop a variety of vaccination strategies, which ultimately converged into worldwide vaccination efforts. These vaccines are administered IM and induce a mostly systemic immune response against SARS-CoV-2. While providing adequate protection against SARS-CoV-2, waning immunity of the population over time and the emergence of SARS-CoV-2 variants exhibiting both increased transmissibility and immune evasion directed the search for vaccination strategies that confer prolonged immunity and improved protection from both infection and disease. As SARS-CoV-2 is a respiratory virus, induction of mucosal immunity is expected to have a beneficial effect. Here, we explored the integration of IN vaccine administration in the VSV–∆G–spike prime and boost regimen, either as a homologous vaccination (IN + IN) or as a boost for IM vaccination (heterologous IM + IN), in comparison to the current IM + IM vaccination strategy. We showed that incorporation of IN vaccination protects K18–hACE2 mice against SARS-CoV-2, reduces the viral load in target organs, induces both systemic and mucosal immunity and ameliorates the disease-associated gene response. We further demonstrate that the induction of mucosal immunity by IN vaccination maintained antibody titers and conferred protection even one year after vaccination. We also showed the prevention of virus transmission in a co-caged golden Syrian hamster model.

We have previously demonstrated the efficacy of a single-dose IM vaccination of VSV–∆G–spike in the golden Syrian hamster model [15]. We showed that this vaccination prevented weight loss, reduced the viral load, prevented tissue damage and induced neutralizing antibodies, as well as a balanced Th1/Th2 response [15]. We also demonstrated the protective efficacy of VSV–∆G–spike prime and boost IM vaccination in the K18–hACE2 mice model against alpha, beta, gamma and delta variants [21] by preventing morbidity and mortality and reducing the viral load in target organs. However, while the IM + IM vaccination nearly eliminated the virus’ presence in the lungs and brains, SARS-CoV-2 was still evident at the nasal turbinates (NTs) at 2 dpi. Residual virus in the NTs, namely the site of infection, may allow virus spreading. Thus, in this current work, we evaluated whether induction of mucosal immunity by IN vaccine administration, either by IM + IN or IN + IN regimen, can reduce the virus in the NTs. Indeed, we showed that mice vaccinated at least once intranasally (IM + IN or IN + IN) had lower viral loads in the NTs; some were below the limit of detection, and the IN + IN regimen was as effective as the IM + IM and IM + IN regimens, even though a two orders of magnitude lower dose was used by this regimen. This lower vaccination dose was as effective also in protecting the mice against the lethal SARS-CoV-2 challenge. We also showed that at 2 dpi, some organs were completely clear of the virus as far as detectable.

Mucosal surfaces serve as a physical barrier but also comprise innate and adaptive immune components, including circulating IgG and localized IgA antibodies [46]. IgA is known to be the most prevalent Ig class at mucosal sites and can also be found in serum [47]. A positive correlation between high levels of anti-influenza IgA and a reduced incidence of influenza illness in vaccinated children has been demonstrated following IN vaccination against the influenza virus [48]. By comparing the IgG and IgA antibodies in the serum and BALFs of the vaccinated mice, we showed that while the IM + IM vaccination induces IgG antibodies, IgA in the serum was only detected upon nasal vaccination by either the IM + IN or IN + IN regimens. Thus, heterologous vaccination (IM + IN) allows for the induction of both systemic and mucosal immunity, as previously documented using other vaccine platforms [12,49,50,51]. When analyzing the antibody repertoire in the BALFs, we found that IgA was preferentially present in the IM + IN-vaccinated mice, while homologous IN + IN nasal vaccination at a significantly lower dose of 2 log was preferentially associated with IgG antibodies in the BALFs. When we analyzed the CD8+ T-cell activity in both the spleen and lungs, we observed a higher rate of T-cell activation and an increased percentage of spike-specific effector memory T-cells in the lungs in mice that were vaccinated by heterologous IM + IN vaccination compared to the homologous vaccination regimens. Whether these observations indicate that, on the background of prior immunity, the primary exposure of the lungs to the vaccine virus (upon IM + IN vaccination) induces a more primary mucosal immune response while a second exposure (following IN + IN vaccination) induces a lung resident memory type of response requires further investigation. The reduced dose of the IN + IN regimen compared to the IM + IN regimen might also contribute to these results.

We previously performed RNA sequencing and pathway analysis to characterize the disease state in infected mice’s brains at 5 days post-infection with SARS-CoV-2 variants and showed robust cytokine storm and neuroinflammation in unvaccinated mice infected with various SARS-CoV-2 variants. Such host response was undetectable in 10^7^ IM + IM-vaccinated mice [21]. COVID-19 disease manifests, among other organs, in central nervous system symptoms [52], and disease severity and death correlate with SARS-CoV-2 viral load in the brain of K-18 hACE2 transgenic mice [33]. Thus, the brain is an important target for SARS-CoV-2 infection. The evaluation of the effect of IN instillation on the brain is of value due to the direct connection between the nasal cavity and the central nervous system (CNS) through the olfactory system [53]. Similar to our previous work, homologous IM + IM vaccination, though administered at a lower dose than before (10^6^ pfu vs. 10^7^ pfu), rarely resulted in an enrichment of DEGs or signaling pathways upon SARS-CoV-2 challenge. The IN + IN vaccination followed by SARS-CoV-2 challenge also resulted in a lower number of DEGs than in the unvaccinated and infected brains, and nearly no enriched pathways were detected in this work. Overall, both homologous vaccination regimens resulted in a brain RNA profile that resembled naïve brains. However, the unvaccinated mice’s brains displayed many DEGs, and the highly enriched DEGs in the unvaccinated group were mainly CCL and CXCL family members. We also showed that at 4 dpi, the unvaccinated mice’s brains displayed a robust enrichment of signaling pathways, many of which fall within several categories of interest: cellular immune response, humoral immune response, cytokine signaling, disease-specific pathways and pathogen-influenced signaling. These categories are further subdivided into dozens of pathways. While the heterologous IM + IN vaccination and infection led to an enrichment of pathways, their significance in the IM + IN group was much lower than in the unvaccinated group. Importantly, whereas the unvaccinated brains displayed an enrichment of pathogenesis pathways and cytokine storm pathways, we showed that the IM + IN vaccination and infection mouse brains mostly displayed pathways that are categorized into cellular immune response category, including “Th1 pathways” and “Th1 and Th2 activation pathways”. The Th1 pathways promote humoral and cell-mediated immune responses associated mostly with virus clearance and protective immunity [54]. Following SARS-CoV-2 infection, Th1 response was shown to be important for viral clearance, whereas an excessive immune response leading to cytokine storm may trigger a Th2 response, which was shown to have a poor prognosis. Following vaccination, there is concern of a Th2-biased response in association with vaccine-associated enhanced respiratory disease [55]. Thus, a balanced Th1/Th2 response is important for both host defense and vaccine safety. It was previously shown that IN prime and boost vaccination with recombinant adenovirus that expresses the spike protein of SARS-CoV-2 led to significantly balanced Th1/Th2 immunity and reduced pulmonary viral loads and inflammation in BALB/c mice [56]. Our previous work evaluated spike-specific IgG2c and IgG1 isotypes as surrogates for Th1 and Th2, respectively, in serum samples of mice vaccinated with a single IM vaccination with VSV–∆G–spike and showed a significant Th1 response [15]. Here, pathway analysis of the IM + IN-vaccinated mice’s brains may further support a balanced Th1/Th2 response.

Prolonged protection against current and future SARS-CoV-2 variants relies on the durability of vaccine-induced immunity. Others have explored long-term protection against SARS-CoV-2 with different vaccination strategies that include IN administration, such as a single dose of an IN adenovirus vector-based vaccine [11] and an IM mRNA-LNP prime vaccine followed by an unadjuvanted spike IN boost [12] and showed long-term protection for as long as 200 days or 118 days post-vaccination, respectively [11,12]. Here, we report that IN inoculation with VSV–∆G–spike has a beneficial effect on long-term immunity and protection, as evident by the detection of anti-spike IgG and IgA antibodies at one year post-vaccination and complete protection following delta-variant SARS-CoV-2 infection.

Airborne transmission models in hamsters to evaluate virus transmission have been previously reported [12,13,45]. Depending on the experimental settings, these experiments allowed the questions of virus transmission through contact and/or airborne transmission to be addressed. In this study, we established a co-caging model for hamsters in which the donor and acceptor compartments were separated by a semi-permeable metal net. Our model is unique in that it allows both uninterrupted air flow and limited contact between the animals for 24 h, mimicking a natural environment.

Knisely et al. recently highlighted the scientific gaps and challenges, as well as the advantages, of mucosal vaccines [6]. In light of these gaps and challenges, we showed that the VSV–∆G–spike-based vaccine induces protective systemic and mucosal long-lasting immunity on the background of preexisting immunity. The worldwide vaccination campaign against SARS-CoV-2 was achieved by a standard IM injection. The potential of IN immunization against SARS-CoV-2 through various vaccines and vaccine candidates has been demonstrated by several studies [6,13,50]. We further investigated the potential of IN vaccination on the background of a previous IM injection and demonstrated its efficacy against lung damage by a respiratory virus, namely SARS-CoV-2. We can assume that an IM + IN regimen for routinely used vaccines against respiratory pathogens such as influenza, measles, *S. pneumoniae* and *B. pertussis* may be advantageous as well and should be evaluated.

This study has several limitations. First, the RNAseq analysis revealed heterogenicity within the unvaccinated group and the IM + IN group. This may account for the observed enriched DEGs and pathways shared by these two groups, though the vaccinated IM + IN group showed significantly fewer enriched DEGs and pathways than the unvaccinated group. The sampling time (brains at 4 dpi) and vaccination efficiency may also account for the differences within each group and the observed transcriptomic profiles. Second, our data on long-term immunity and protection following IN vaccination are based on a small sample size and a limited set of vaccination doses that differ from the doses chosen for the short-term experiments. Due to the preliminary nature of these results, the relatively small sample size and the narrow range of the tested doses, these findings should be further explored in future work. Third, our hamster co-caging experiments mainly focused on the heterologous IM + IN route. It would be interesting to compare these data to the homologous regimens, especially the IN + IN regimen, and also to test uniform vaccination doses. However, this is beyond the scope of the current work. Forth, currently, omicron sublineages are the predominant SARS-CoV-2 variants, and their virulence in K18–hACE2 mice varies. For the current study, in order to demonstrate the advantages of the intranasal vaccination, we chose the delta variant, as we had previously established its ability to cause severe and lethal disease [20]. In future work, the efficacy towards additional variants should be tested.

## 5. Conclusions

In conclusion, we utilized K18–hACE2 mouse and golden Syrian hamster models, two well-established SARS-CoV-2 models that complement each other, to show that IN instillation using a VSV–∆G–spike-based vaccine can induce long-term (1 year) mucosal and systemic protective immunity and prevent virus transmission. Moreover, our data support a heterologous IM + IN regimen, mimicking the vaccination strategy of administering an IN boost to a previously IM-vaccinated population, as an effective vaccination strategy against SARS-CoV-2.

## Figures and Tables

**Figure 1 vaccines-12-00491-f001:**
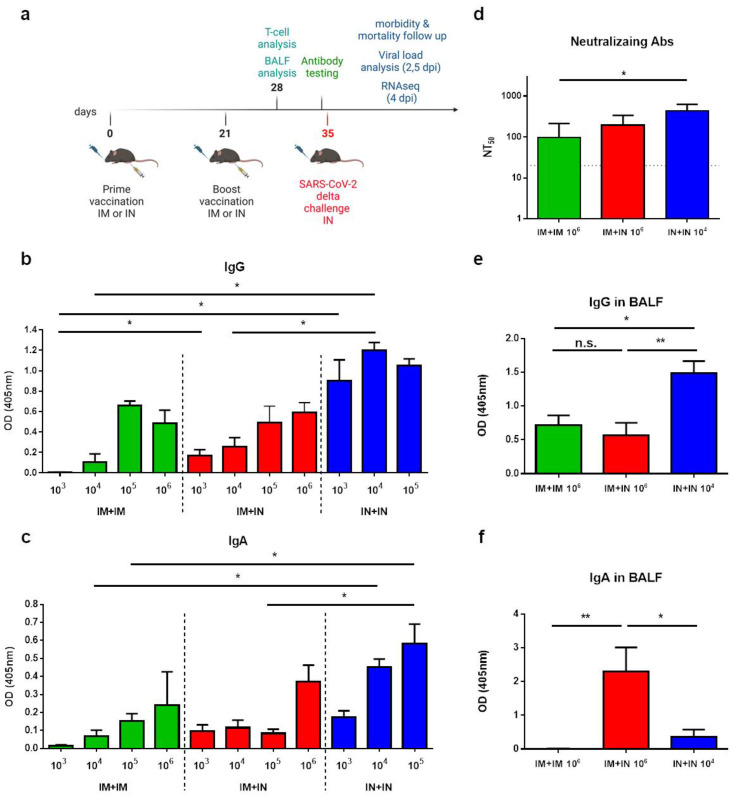
Induction of humoral immunity following mucosal or muscular vaccination: (**a**) Schematic drawing depicting the K18–hACE2 mouse study design: K18–hACE2 mice were vaccinated primarily via either an IM or IN route with the VSV–∆G–spike vaccine, followed by a boost vaccination via either an IM or IN route at 3-week intervals. One week following the boost vaccination, T-cells and BALF analysis were performed. Serum was collected 14 days post boostvaccination. Mice were infected IN with the delta variant of SARS-CoV-2 14 days post boost-vaccination, then monitored daily for morbidity and mortality and subjected to further analyses. The scheme was created with BioRender.com (full license). (**b**) The induction of anti-spike IgG by 1 of 3 vaccination routes, IM + IM, IM + IN or IN + IN, at various doses (pfu/mouse). Each color represents a route of administration. (**c**) The induction of anti-spike IgA by 1 of 3 vaccination routes, IM + IM, IM + IN or IN + IN, at various doses (pfu/mouse). Each color represents a route of administration. Data are presented as the mean ± SEM. For panels (**b**,**c**), n = 8 for IN + IN groups, n = 4 for IM + IN or IM + IM groups. For panels (**b**,**c**), analysis was performed using a one-way ANOVA with Tukey’s post hoc test: * *p* < 0.05; ** *p* < 0.01. Matched doses are statistically compared in panels (**b**,**c**). Full statistical analysis is presented in Appendix A, respectively. (**d**) The induction of neutralizing antibodies by IM + IM vaccination (n = 6), IM + IN vaccination (n = 4), or IN + IN vaccination (n = 4). (**e**,**f**) ELISA of IgG and IgA levels in the BALFs of mice vaccinated with 1 of 3 vaccination routes: IM + IM (n = 6), IM + IN (n = 5), IN + IN (n = 5). Doses are pfu/mouse. Statistical analysis was performed using a one-way ANOVA with Tukey’s post hoc test: * *p* < 0.05; ** *p* < 0.01; n.s. = not significant.

**Figure 2 vaccines-12-00491-f002:**
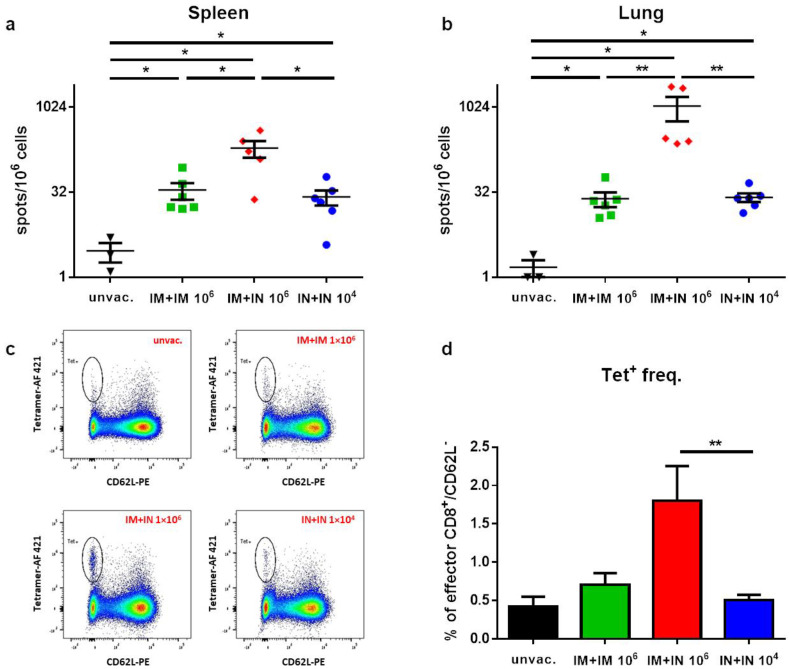
Induction of CD8+ T-cell response following different vaccination regimens with VSV–ΔG–spike: Seven days post 2nd immunization, spike-specific T-cells in the (**a**) spleen and (**b**) lungs were enumerated by ELISpot assay. (**c**) Frequencies of spike-specific effector CD8+ T-cells in the lungs were determined by flow cytometry. A representative analysis from each group is shown. (**d**) Histograms incorporating the individual sets of results obtained for each animal. Tet+ stands for tetramer. Gating strategy for c appears in Appendix A. Bars indicate means ± SEM from 3 to 6 animals per group. Statistical analysis was performed by Mann–Whitney nonparametric *t*-test. * *p* < 0.05; ** *p* < 0.01.

**Figure 3 vaccines-12-00491-f003:**
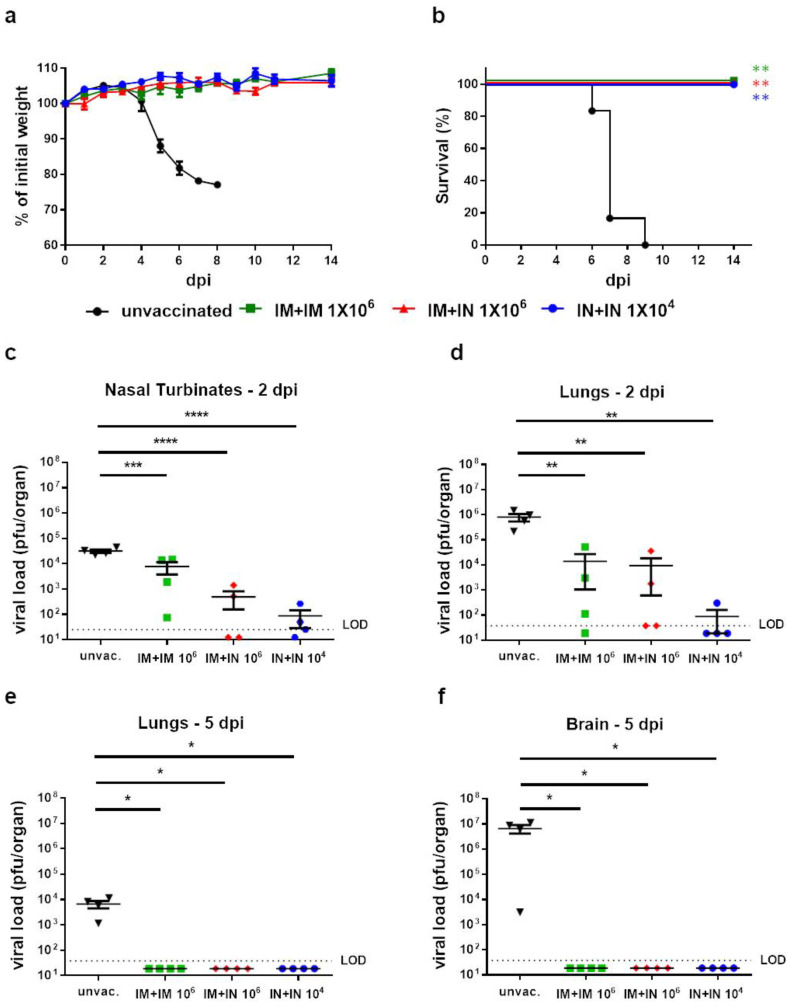
Protection of K18–hACE2 mice by all tested routes of vaccination: (**a**) Body weight changes and (**b**) survival curve of K18–hACE2 mice vaccinated by homologous IM, homologous IN or heterologous IM + IN vaccination of VSV–ΔG–spike or unvaccinated mice following delta variant SARS-CoV-2 infection (10,000 pfu). n = 6 for the IM + IN, IN + IN and unvaccinated groups, and n = 5 for the IM + IM group. Statistical analysis was performed using log-rank (Mantel–Cox) test: ** *p* < 0.01. (**c**–**f**) Viral load analysis of unvaccinated K18–hACE2 mice or K18–hACE2 mice vaccinated by homologous IM vaccination, homologous IN vaccination, or heterologous IM + IN vaccination of VSV–∆G–spike, following infection with the delta variant of SARS-CoV-2 (10,000 pfu). (**c**) Viral titers at 2 dpi in nasal turbinates (NTs), (**d**) viral titers at 2 dpi in lungs, (**e**) viral titers at 5 dpi in lungs, and (**f**) viral titers at 5 dpi in brains. Viral titers were determined by plaque assay and are presented as pfu/organ. Data are presented as means ± SEM. For each organ at each time point, n = 4. Values below the limit of detection (LOD) were assigned values of half LOD. Statistical analysis was performed using a one-way ANOVA with Tukey’s post hoc test. * *p* < 0.05; ** *p* < 0.01; *** *p* < 0.001; **** *p* < 0.0001.

**Figure 4 vaccines-12-00491-f004:**
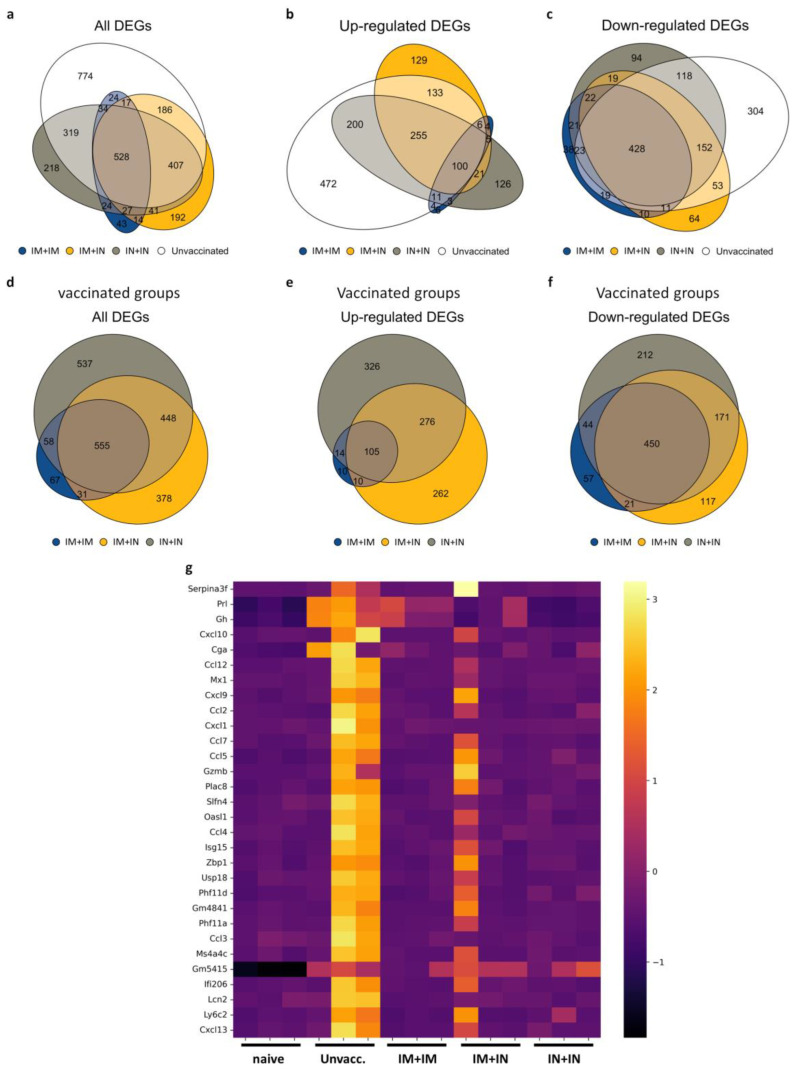
Mucosal vaccination affects brain gene expression patterns following delta SARS-CoV-2 infection**:** (**a**–**c**) Area-proportional Venn diagrams of overlapping genes of unvaccinated, IM + IM-vaccinated, IM + IN-vaccinated and IN + IN-vaccinated mice brains at 4 days post-infection with the delta variant of SARS-CoV-2, showing (**a**) all DEGs, (**b**) upregulated DEGs and (**c**) downregulated DEGs. (**d**–**f**) Area-proportional Venn diagrams of overlapping genes in vaccinated groups, IM + IM, IM + IN and IN + IN mouse brains, at 4 days post-infection with the delta variant of SARS-CoV-2, showing (**d**) all vaccinated groups’ DEGs, (**e**) upregulated DEGs of vaccinated groups and (**f**) downregulated DEGs of vaccinated groups. n = 3 for each group. Log2 Fold Change (Log2FC) > 1 or <−1 for upregulated or downregulated DEGs, respectively. Each individual group is compared to naïve mouse brains prior to comparison to other groups. padj ≤ 0.05. (**g**) Heatmap of selected DEGs in all tested groups. Top 30 genes were selected based on the unvaccinated group’s Fold Change score. Scale represents z-score obtained on normalized reads. Adjusted *p*-value < 0.05.

**Figure 5 vaccines-12-00491-f005:**
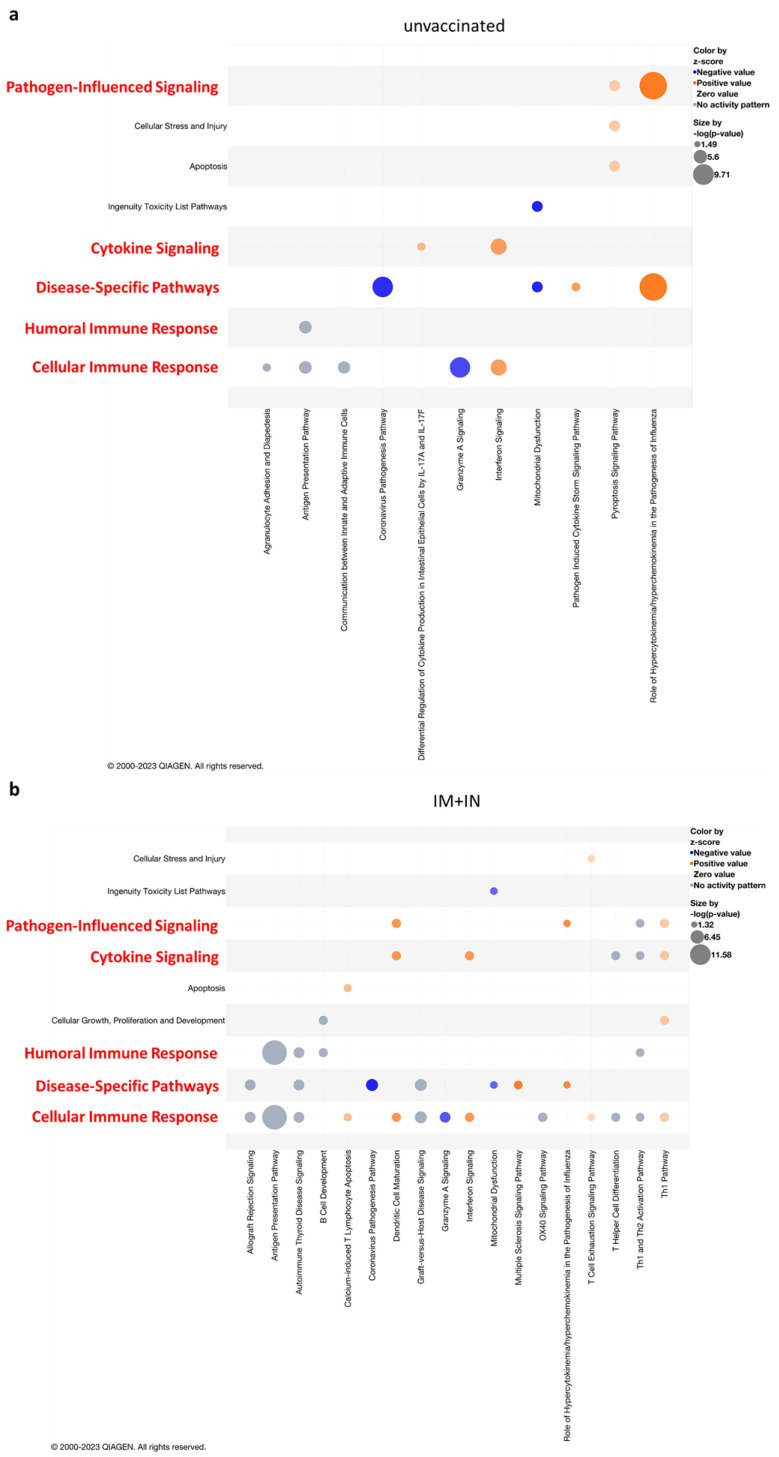
Reduced enrichment of cellular and humoral immune responses, cytokine signaling and disease− and pathogen−influenced signaling by mucosal vaccination. Ingenuity-enriched pathway analysis bubble plots for each group: (**a**) unvaccinated group and (**b**) IM + IN-vaccinated K18–hACE2 mice’s brains at 4 days post-delta-variant SARS-CoV-2 infection. Each bubble plot shows enriched pathways arranged by categories of interest: cellular immune response, cytokine signaling, disease−specific pathways, humoral immune response and pathogen−influenced signaling (*y*-axis, highlighted in red), and the enriched canonical pathways that map to each category (*x*-axis), arranged in an alphabetical order. Color by z-score: orange indicates pathway activation, blue indicates pathway inhibition and grey represents no activity pattern, as predicted by Ingenuity knowledge. Bubble size represents -log (*p*-value). Legend appears on the upper right side of each panel. Pathway enrichment was set to a cutoff of <0.05. n = 3 for each group. Right-tailed Fisher’s exact test. A Benjamini–Hochberg multiple-comparisons correction test was applied.

**Figure 6 vaccines-12-00491-f006:**
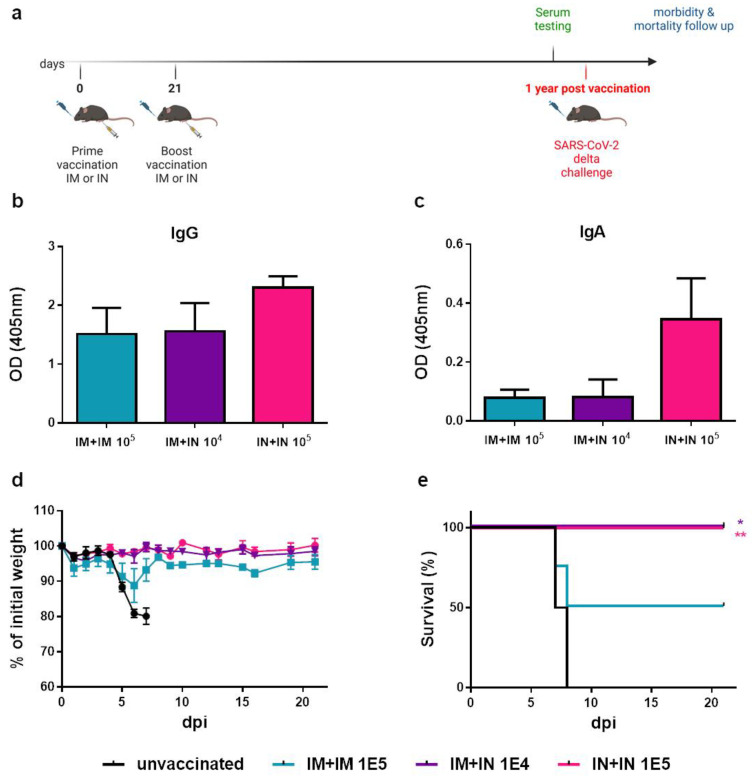
Long-lasting immunity of K18–hACE2 mice one year following IN instillation with VSV–ΔG–spike: (**a**) Schematic drawing of longevity timeline. Scheme was created with BioRender.com (full license). (**b**) Anti-spike IgG levels and (**c**) anti-spike IgA levels in serum samples of K18–hACE2 mice one year following boost vaccination with VSV–ΔG–spike. (**d**) Body weight monitoring and (**e**) survival curve of K18–hACE2 mice following infection with the delta variant of SARS-CoV-2. Mice were either unvaccinated or following vaccination by homologous IM, homologous IN, or heterologous IM + IN vaccination one year prior to infection. Statistical analysis was performed using a log-rank (Mantel–Cox) test: * *p* < 0.05; ** *p* < 0.01. n = 4 for all groups, except the IM + IN 10^4^ group (n = 3).

**Figure 7 vaccines-12-00491-f007:**
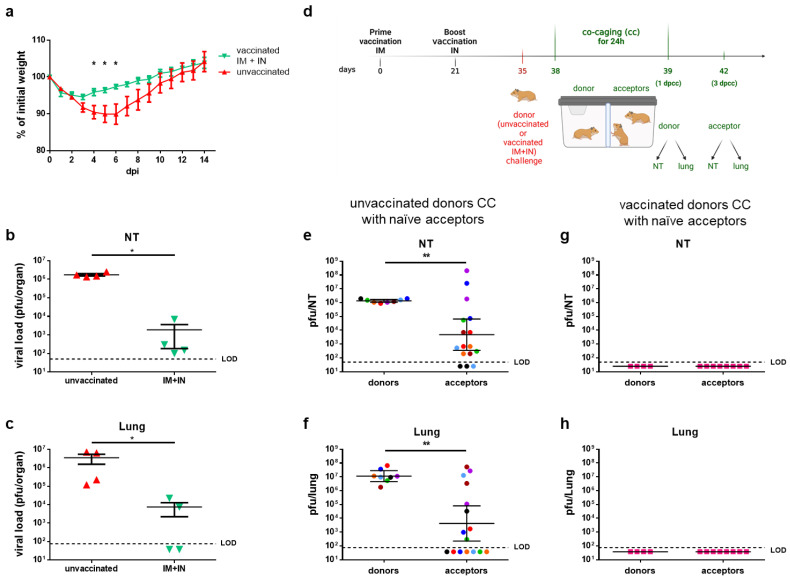
Heterologous vaccination of golden Syrian hamsters prevents SARS-CoV-2 virus transmission: (**a**) Body weight changes of unvaccinated hamsters or hamsters vaccinated with heterologous IM + IN injection of VSV–ΔG–spike. The scheme was created with BioRender.com (full license). (**b**,**c**) Viral load titers at 3 dpi in the (**b**) nasal turbinates (NTs) or (**c**) lungs of unvaccinated hamsters or hamsters following heterologous IM + IN vaccination. n = 4 for each group. (**d**) Schematic drawing displaying co-caging (“cc”) setting of unvaccinated or vaccinated donors and naïve hamsters following the SARS-CoV-2 infection of donor hamsters. “dpcc” stands for “days post-co-caging”. (**e**,**f**) Viral load titers in the (**e**) NTs or (**f**) lungs of unvaccinated donors and naïve acceptors following co-caging. (**g**,**h**) Viral load titers in the (**g**) nasal turbinates or (**h**) lungs of IM + IN-vaccinated donors and naïve acceptors following co-caging. Donors: n = 8; acceptors: n = 16. Each color represents a set of 1 donor and 2 acceptors co-caged. Statistical analysis was performed by a Mann–Whitney nonparametric *t*-test. * *p* < 0.05; ** *p* < 0.01.

## Data Availability

The raw data supporting the conclusions of this article will be made available by authors on request.

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
