# Peer review of "Induction of Superior Systemic and Mucosal Protective Immunity to SARS-CoV-2 by Nasal Administration of a VSV–ΔG–Spike Vaccine"

_vaccines, 2024, doi:10.3390/vaccines12050491_

Round 1
Reviewer 1 Report
Comments and Suggestions for Authors
The rapid spread of SARS-CoV-2 variants poses a significant challenge to the efficacy of existing vaccines. While intramuscular (IM) administration provides short-lived immunity, it fails to prevent infection and transmission effectively. Therefore, novel vaccination strategies are imperative to extend vaccine effectiveness. In their study, Yahalom-Ronen and colleagues demonstrated that intranasal (IN) delivery of the VSV-ΔG-Spike vaccine directly to mucosal surfaces induced robust mucosal and systemic immune responses. Compared to IM vaccination in the K18-hACE2 model, IN vaccination predominantly stimulated mucosal IgA and T-cell responses, decreased viral load at the infection site, and mitigated disease-associated brain gene expression.
Importantly, IN vaccination maintained its protective effect even one year post-administration. These findings underscore the potential of a heterologous IM+IN vaccination regimen to elicit mucosal immunity alongside systemic protection. Moreover, the IM+IN regimen showed efficacy in preventing virus transmission in a golden Syrian hamster co-caging model.
In summary, this study highlights the favorable potential of IN vaccination with VSV-ΔG-Spike, either as a homologous IN+IN regimen or as a booster following IM vaccination.
Overall, this preclinical trial of a vaccine candidate presents strong results supporting the use of the VSV-ΔG-Spike vaccine candidate via the IN route. However, it is essential to note that this proof of concept requires further evaluation to assess its efficacy against SARS-CoV-2 Omicron variants. Since the Omicron variant emerged towards the end of 2021 and has been the predominant lineage circulating globally since early 2022, it is crucial to conduct experiments to evaluate the vaccine candidate's potential efficacy against this current variant.
I strongly recommend that the authors conduct a challenging study to assess protection using a current isolate of the Omicron sublineage. Alternatively, the discussion should address the limitations of the results since the Delta variant of concern has not been circulating since 2022. This should be included in the paragraph describing the study's limitations.
Reviewer 2 Report
Comments and Suggestions for Authors
This study applied Vesicular Stomatitis Virus (VSV) to carry SARS-CoV-2 Spike protein for intranasal (IN) and/or intramuscular (IM) vaccination and booster in mice and hamsters, and characterized titers of neutralizing antibodies, protection efficacy from SARS-CoV-2 infection, and differentially expressed genes. Although the use of VSV for SARS-CoV-2 vaccination has been reported before, this study demonstrated intranasal initial and booster vaccination, even at low VSV titer, works better than IM+IN and IM+IM vaccination regimens. The IN+IN showed higher protection from SARS-CoV-2 infection and beneficial cytokine and gene expression profiles than other combinations.
However, VSV-wild-type control was not included. The effect of VSV alone cannot be determined and a possible contribution of VSV alone to the observed effects cannot be excluded. Also, the pathology and possible adverse effects were not reported in these animal models. These are critical for future study in primate or in clinical trial.
Round 2
Reviewer 2 Report
Comments and Suggestions for Authors
The revised manuscript is satisfactory.